# The Feasibility and Preliminary Efficacy of an eHealth Lifestyle Program in Women with Recent Gestational Diabetes Mellitus: A Pilot Study

**DOI:** 10.3390/ijerph17197115

**Published:** 2020-09-28

**Authors:** Megan E. Rollo, Jennifer N. Baldwin, Melinda Hutchesson, Elroy J. Aguiar, Katie Wynne, Ashley Young, Robin Callister, Rebecca Haslam, Clare E. Collins

**Affiliations:** 1School of Health Sciences, Faculty of Health and Medicine, University of Newcastle, Callaghan, NSW 2308, Australia; Jennifer.Baldwin@newcastle.edu.au (J.N.B.); melinda.hutchesson@newcastle.edu.au (M.H.); Rebecca.Williams@newcastle.edu.au (R.H.); 2Priority Research Centre for Physical Activity and Nutrition, University of Newcastle, Callaghan, NSW 2308, Australia; robin.callister@newcastle.edu.au; 3Department of Kinesiology, College of Education, The University of Alabama, Tuscaloosa, AL 35487, USA; ejaguiar@ua.edu; 4School of Medicine and Public Health, Faculty of Health and Medicine, University of Newcastle, Callaghan, NSW 2308, Australia; katie-jane.wynne@newcastle.edu.au; 5Department of Diabetes & Endocrinology, John Hunter Hospital, Hunter New England Health District, New Lambton Heights, NSW 2305, Australia; 6Clinical Services, Nursing and Midwifery, Hunter New England Local Health District, Wallsend, NSW 2287, Australia; Ashley.Young@health.nsw.gov.au; 7School of Biomedical Sciences and Pharmacy, Faculty of Health and Medicine, University of Newcastle, Callaghan, NSW 2308, Australia

**Keywords:** gestational diabetes, type 2 diabetes mellitus, prevention, weight loss, diet, exercise

## Abstract

Self-administered eHealth interventions provide a potential low-cost solution for reducing diabetes risk. The aim of this pilot randomised controlled trial (RCT) was to evaluate the feasibility, including recruitment, retention, preliminary efficacy (primary outcome) and acceptability (secondary outcome) of the “Body Balance Beyond” eHealth intervention in women with previous gestational diabetes mellitus (GDM). Women with overweight/obesity who had recent GDM (previous 24 months) were randomised into one of three groups: 1) high personalisation (access to “Body Balance Beyond” website, individual telehealth coaching via video call by a dietitian and exercise physiologist, and text message support); 2) low personalisation (website only); or 3) waitlist control. To evaluate preliminary efficacy, weight (kg), glycosylated hemoglobin, type A1C (HbA1c), cholesterol (total, low-density lipoprotein (LDL) and high-density lipoprotein (HDL)), diet quality and moderate–vigorous physical activity were analysed at baseline and at 3 and 6 months using generalised linear mixed models. To investigate acceptability, process evaluation was conducted at 3 and 6 months. Of the 327 potential participants screened, 42 women (mean age 33.5 ± 4.0 years and BMI 32.4 ± 4.3 kg/m^2^) were randomised, with 30 (71%) completing the study. Retention at 6 months was 80%, 54% and 79% for high personalisation, low personalisation and waitlist control, respectively (reasons: personal/work commitments, *n* = 4; started weight-loss diet, *n* = 1; pregnant, *n* = 1; resources not useful, *n* = 1; and not contactable, *n* = 5). No significant group-by-time interactions were observed for preliminary efficacy outcomes, with the exception of HDL cholesterol, where a difference favoured the low personalisation group relative to the control (*p* = 0.028). The majority (91%) of women accessed the website in the first 3 months and 57% from 4–6 months. The website provided useful information for 95% and 92% of women at 3 and 6 months, respectively, although only a third of women found it motivating (30% and 25% at 3 and 6 months, respectively). Most women agreed that the telehealth coaching increased their confidence for improving diet (85%) and physical activity (92%) behaviours, although fewer women regarded the text messages as positive (22% and 31% for improving diet and physical activity, respectively). The majority of women (82% at 3 months and 87% at 6 months) in the high personalisation group would recommend the program to other women with GDM. Recruiting and retaining women with a recent diagnosis of GDM is challenging. The “Body Balance Beyond” website combined with telehealth coaching via video call is largely acceptable and useful for women with recent GDM. Further analysis of the effect on diabetes risk reduction in a larger study is needed.

## 1. Introduction

Type 2 diabetes mellitus (T2DM) prevalence is escalating globally. An estimated 422 million people had diabetes in 2014, of which approximately 85% had T2DM [1]. Gestational diabetes mellitus (GDM), defined as glucose intolerance diagnosed for the first time during pregnancy [2], occurs in approximately 5–15% of pregnant women in Australia [3,4,5]. In recent decades, the prevalence of GDM has increased by more than 30% in several countries, including developing countries [6,7]. Women with a history of GDM have a 9-fold increased risk of developing future T2DM [8].

Lifestyle interventions, typically involving physical activity, nutrition and education components to achieve weight loss, are the main strategies for T2DM risk reduction for women with a history of GDM [9,10]. Lifestyle interventions can improve diabetes risk factor levels in women with previous GDM [9,11,12,13], although the effect on development of T2DM for up to 10 years post-GDM is unclear [9]. Implementing lifestyle interventions for women with a history of GDM can be challenging due to factors such as risk perception, health beliefs, social support and self-efficacy [14,15,16].

Delivering lifestyle interventions can be a particular challenge for women with young children. Electronic health (eHealth) technologies provide a potential solution for supporting diabetes self-management [17] Interventions delivered using eHealth technologies have the potential to address common barriers faced by postpartum women, such as time and competing demands, by providing a flexible method of delivery that promotes sustained engagement and self-monitoring [18,19]. A systematic review found that eHealth interventions had a positive effect on weight management compared to control among postpartum women [20]. Among women with a history of GDM, face-to-face and telephone counselling improved psychosocial factors related to physical activity and nutrition [21], whereas a web-based pedometer program had no effect on weight, behavioural constructs or fasting plasma glucose [22]. Pilot studies have shown preliminary positive impacts on lifestyle behaviours following a telephone motivational interviewing program in women with GDM [23] and on knowledge of GDM after a web-based intervention [24]. Further research into the effects of eHealth technologies on diabetes risk reduction in women with a history of GDM is needed.

The “Body Balance Beyond” program is a 6-month eHealth, self-directed lifestyle program comprising a healthy lifestyle website with individualised telehealth video coaching and text message support. The program is designed to promote modest weight loss by supporting participants to form healthy eating and physical activity behaviours. The aim of this pilot study was to evaluate the feasibility of the “Body Balance Beyond” eHealth intervention in women with recent GDM, with a view to inform a larger randomised controlled trial. Key objectives relating to this aim were 1) assessing recruitment and retention of women with recent GDM; 2) investigating preliminary impact of the intervention on weight, HbA1c, cholesterol (total, low-density lipoprotein (LDL) and high-density lipoprotein (HDL)), diet quality and moderate–vigorous physical activity; and 3) evaluating intervention acceptability including satisfaction, usability, appropriateness and usage.

## 2. Materials and Methods

### 2.1. Study Design

This study was a 6-month, three-arm pilot randomised controlled trial (RCT) for women with recent GDM who were at risk of developing T2DM. The study protocol is summarized in Figure 1. Ethical approval for this study was granted by the institutional Human Research Ethics Committee (approval number H-2017-0187). The study was registered with the Australian New Zealand Clinical Trials Registry (ANZCTR): ACTRN12617001456347. The design, conduct and reporting of this study were in accordance with the Consolidated Standards of Reporting Trials (CONSORT) guidelines [25]. A process evaluation was embedded within the study to assess acceptability and feasibility of the intervention.

### 2.2. Participants

Eligible participants were women aged 18–45 years who had been diagnosed with GDM in the past 24 months, were greater than 3 months postpartum, and with overweight or obesity (body mass index (BMI) 25–50 kg/m^2^. Exclusion criteria were currently pregnant or trying to fall pregnant (includes women who became pregnant during the course of the study), no medical clearance to exercise, medical condition or injury that could be exacerbated by exercise, diagnosed with type 1 or type 2 diabetes mellitus, home internet download/upload speed <0.3 Mbps, screening questionnaire not completed and unable to attend assessment sessions.

Participants were recruited from the Hunter and Central Coast regions of New South Wales, Australia using two main strategies: (1) media releases and social media posts, and (2) invitation letters posted to women with previous GDM and a residential postcode in the above areas who were registered on the Diabetes Australia National Diabetes Supply Scheme Register.

Interested participants contacted the research team via email or phone and were directed to an online screening questionnaire to assess eligibility. BMI was assessed using each participant’s self-reported height and weight. Potential participants who reported having medical issues affecting their ability to exercise were required to seek clearance from their medical practitioner in order to be eligible to participate. Upon completion of the screening questionnaire, eligible participants were asked to provide their contact details and were then emailed an information statement outlining the requirements of participating in the study, potential benefits and risks, and a consent form. All participants provided written informed consent.

As this was a pilot study, a power calculation was not performed to determine sample size [26]. A pragmatic target of 90 participants (30 in each group) was set prior to the study.

### 2.3. Study Groups

Eligible participants were randomised (block list randomisation, block size 6; Sealed Envelope Ltd., London, UK, 2020, https://www.sealedenvelope.com) to one of three groups: high personalisation, low personalisation or waitlist control.

#### 2.3.1. High Personalisation Group

Participants received access to a healthy lifestyle web program, “Body Balance Beyond”, tailored to women with a history of GDM. The materials on the website were predominantly self-directed and designed to promote modest weight loss by supporting participants to make a series of small changes to eating and exercise behaviours and to form new healthy behaviours. Participants had access to the website for 6 months and could access the website at their discretion.

The website had five content sections:Managing my risk: outlined the risk factors for developing diabetes and actions that can reduce risk.My Plan: participants were emailed personalised feedback reports on their dietary intake and physical activity (generated from the Australian Eating Survey [27] and Godin Leisure-time Exercise Questionnaire [28,29]) compared to national nutrition and exercise guidelines. This section guided participants through using their personal reports to set weight, diet and exercise-related goals and to develop strategies for self-monitoring and managing relapses.Eating: information and links to resources to promote healthy eating, such as portion size guidelines, energy density, meal planning, reading food labels, recipes and eating when breastfeeding.Physical Activity: information and links to resources to promote being more active, including different types of physical activity, benefits of being physically active, suggestions for being active with their family and exercising safely.Wellbeing: information and resources on social support, stress management and sleep.

In addition, participants in the high personalisation group were provided with six individual telehealth coaching sessions delivered via video call (20–30 min duration each) with a dietitian (weeks 2, 5 and 9) and exercise physiologist (weeks 3, 6 and 10) over the first 3 months. The coaching sessions were delivered using the Scopia© platform (Version 8.3.900, Avaya Inc., Santa Clara, CA, USA) and incorporated some learnings from the delivery of video coaching sessions in a previous study in postpartum women which focused on weight management [30,31]. The coaching sessions in the current study consisted of structured participant-centred sessions to review participant dietary intake and exercise levels, to explore barriers to healthy eating and physical activity, to refine participant goals and to provide personalised strategies to support participants in achieving goals. During the initial coaching sessions with the dietitian and exercise physiologist, the practitioners used the responses to the Personalised Nutrition and Personalised Exercise Questionnaires to streamline the selection of strategies to overcome participant’s self-identified factors affecting diet and exercise. The generic capability, opportunity, and motivation-behaviour system (COM-B) questionnaire outlined by Michie, van Stalen and West [32] was adapted for known barriers relating to healthy eating and exercise impacting females with young families. The subsequent coaching sessions were used to review progress, to revise goals, to problem-solve barriers and to revise strategies, as required.

Text message support was provided over the second 3 months. Support offered was based on self-identification of factors perceived to impact the participant’s ability to eat healthy and to be physically active, collected via a subsequent Personalised Nutrition Questionnaire and Personalised Exercise Questionnaire administered at 3 months. When completing the second personalized nutrition and exercise questionnaires, participants prioritized their top 3 nutrition and top 3 exercise factors, and four messages for each factor were sent over 12 weeks. An example text message for the nutrition capability factor of “having the ability to resist food cravings” was “[First name], when cravings hit, have a glass of water and wait a few mins. You may be confusing thirst for hunger. The craving may fade once you have rehydrated.” An example text message for the exercise opportunity factor of “having better access to exercise, physical activity facilities or equipment” was “[First name], exercise equipment is all around you. Here is a park bench circuit [LINK] for you to try, with step ups, tricep dips and many more!” A total of three text messages per week were sent, including one consisting of reminders to self-monitor or reflect on goals and two comprising personalised messages (one diet-related and one exercise-related).

#### 2.3.2. Low Personalisation Group

Participants in the low personalisation group were provided with access to the “Body Balance Beyond” web program for 6 months (as described above).

#### 2.3.3. Waitlist Control Group

Participants were asked to not make any changes to their current diet or exercise and to not attempt to lose weight during the intervention phase. This group was provided with the high personalisation intervention at the end of the active intervention phase (6 months).

### 2.4. Preliminary Efficacy

Assessments were conducted at baseline, 3 months (mid-program) and 6 months (immediately post-program) in the Human Nutrition Laboratory, University of Newcastle, Australia. A standardised assessment protocol was used to collect all measures. Assessors were trained prior to conducting assessments and were blinded to group allocation at all time points. Participants who initially failed to attend their assessment session were contacted by phone and/or email to reschedule their appointment where possible.

#### 2.4.1. Anthropometric Measures

Weight change (kg) was assessed as the primary outcome in this pilot study as weight loss has been identified as the main predictor of reduced diabetes incidence [33]. Weight was measured with the participant wearing light clothing and without shoes using a calibrated digital scale (BSM370, InBody Co., Ltd., Seoul, Korea). Two measures were recorded to the nearest 0.1 kg, and a third measure was obtained if the first and second measures differed by >0.1 kg. The average of the two acceptable measures was reported.

Height (cm) was measured without shoes using an automatic stadiometer (BSM370, InBody Co., Ltd., Seoul, Korea). Two measures were recorded to the nearest 0.1 cm, and a third measure was obtained if the first and second measures differed by >0.3 cm. The average of the two acceptable measures was reported. Height was measured at baseline only. BMI was calculated as weight (kg)/height (m^2^).

Waist circumference (cm) was measured at the midpoint between the lower costal border and the iliac crest, and hip circumference was measured at the level of the greater trochanter. Two measures were recorded to the nearest 0.1 cm, with a third measure taken if there was a difference of >0.5 cm between the first two measures.

Body composition was examined using bioimpedance analysis (InBody720, InBody Co., Ltd., Seoul, Korea) to measure body fat mass (kg) and skeletal muscle mass (kg).

#### 2.4.2. Biochemistry Measures

Blood samples were collected following an overnight fast (minimum 8 h) by a trained phlebotomist using a standardised procedure, and analyses were conducted using standard automated techniques (National Association of Testing Authorities accredited pathology service). The following blood biomarkers were analysed: HbA1c (% and mmol/L), fasting plasma glucose (mmol/L), insulin (mU/L), total cholesterol (mmol/L), LDL cholesterol (mmol/L), HDL cholesterol (mmol/L) and total/HDL cholesterol ratio. Values for glucose and insulin were used to calculate the Homeostatic Model Assessment-2 (HOMA-IR 2) and Quantitative Insulin Sensitivity Check Index (QUICKI) [34,35].

#### 2.4.3. Cardiovascular Measures

Blood pressure was assessed using Pulsecor Cardioscope II (Pulsecor Ltd., Auckland, New Zealand) following a standardised procedure. Participants were asked to sit quietly for 5 min prior to the first measurement, and a 2-min rest was provided before subsequent measures. A minimum of two systolic and diastolic blood pressures were collected (maximum five measures), with acceptable values within the range of 10 mmHg (systolic) or 5 mmHg (diastolic). The mean of the two closest measures were reported.

#### 2.4.4. Diet Quality

Dietary intake was assessed using a validated food frequency questionnaire: the Australian Eating Survey (AES) [27]. The Australian Eating Survey consists of a 120-item semiquantitative food frequency questionnaire with an additional 15 demographic and behavioural questions. Portion sizes were calculated for individual food items using data from the Australia Bureau of Statistics National Nutrition Survey [36] or the “natural” serving size of specific foods where appropriate (e.g., a slice of bread). The frequency of consumption (e.g., “never” up to “4 or more times per day” for most food items and up to “7 or more glasses per day” for beverages) of various food items or types over the previous six months was collected. The AUSNUT 1999 food composition database was used to compute nutrient intakes [37].

Diet quality was measured using a validated brief diet quality index: the Australian Recommended Food Score (ARFS) [38,39]. The ARFS uses a subset of 70 questions from the Australian Eating Survey food frequency questionnaire that aligns with intake of nutrient-dense core foods recommended in the Australian Dietary Guidelines [40]. The ARFS score is calculated by summing the points related to usual weekly variety within eight subscales, with 20 questions related to vegetable intake, 12 related to fruit, 13 related to key protein rich foods (7 to meat and 6 to vegetarian protein sources), 12 to breads/cereals, 10 to dairy and calcium rich foods, one to water and 2 to spreads/sauces. The total score ranges from zero to a maximum of 73 points with further details published elsewhere [38]. The distribution of total energy intake contributed by specific foods was also categorised into two main groups—nutrient-dense, core food groups (e.g., fruits, vegetables, grains) and energy-dense, nutrient-poor, non-core groups (e.g., sweetened drinks, confectionary, takeaway foods)—and subgroups within each of these.

#### 2.4.5. Physical Activity Level

Self-reported physical activity level was assessed using a modified version of the Godin Leisure-time Exercise Questionnaire [28,29] Participants were asked about the average frequency (per week) and average duration (minimum 10 min) of light, moderate and vigorous intensity physical activities they had engaged in over the past month. Responses for frequency and duration for each intensity category were multiplied to generate a measure of the total time spent per week in light, moderate and vigorous physical activity. The total times for moderate and vigorous activities were summed to provide a measure of moderate to vigorous physical activity (minutes/week). Additional questions asked participants about the average weekly frequency and duration of resistance exercise and pelvic floor exercises they had engaged in over the past month.

#### 2.4.6. Self-Efficacy and Quality of Life

Eating self-efficacy was assessed using the Weight Efficacy Lifestyle Questionnaire-Short Form (WEL-SF), an 8-item scale assessing participants’ self-reported confidence in their ability to resist overeating in different settings [41]. Each item was scored on an 11-point scale from “not confident at all” to “very confident”, with scores for individual items summed to give a total score (maximum 90). The WEL-SF has demonstrated acceptable reliability (Cronbach’s alpha = 0.92) and construct validity in a sample of patients with obesity [41,42].

Exercise self-efficacy was assessed using the Self-Efficacy for Exercise scale (SEE), a 9-item questionnaire that asks participants about their perceived confidence in their ability to exercise under different circumstances [43]. Each item was scored on an 11-point scale from “not confident at all” to “very confident”. The SEE was scored by summing the numerical rating for each item and by dividing by the number of responses. The SEE has demonstrated acceptable reliability (Cronbach’s alpha = 0.92) and construct validity [43].

Quality of life was assessed using the Assessment of Quality of Life 6-dimension instrument (AQoL-6D) [44]. The AQoL-6D is a health-related multi-attribute utility instrument that assesses quality of life over six dimensions, generating a global “utility” score reflecting overall quality of life. The AQoL-6D has demonstrated acceptable construct validity, criterion validity and test–retest reliability [44,45].

### 2.5. Acceptability (Secondary Outcome)

A process evaluation survey was included in the study design to examine the acceptability of the program in terms of satisfaction, usability, appropriateness and usage at 3 and 6 months. Participants in both intervention groups were asked about their experience of and engagement with different aspects of the intervention (high personalisation: “Body Balance Beyond” website and goal-setting module, AES and physical activity reports, telehealth video coaching sessions and text messages; low personalisation: “Body Balance Beyond” website and goal-setting module, and AES and physical activity reports).

Satisfaction: Participants were asked to indicate their level of satisfaction with each program component (five-point Likert scale, very unsatisfied = 1, very satisfied = 5).

Usability: Participants were asked to rate the usability (e.g., “was easy to navigate”) and ability to engage (e.g., “made me feel accountable”) aspects of the “Body Balance Beyond” website, telehealth video coaching, AES and physical activity reports, and text messages on a five-point Likert scale (strongly disagree = 1 and strongly agree = 5). Participants were also asked to rate the attractiveness of the website (e.g., “was visually appealing”) and their experience of the coaching sessions (e.g., “the picture quality of the video coaching sessions was acceptable”), also on a five-point Likert scale. Participants were asked whether they had any technology-related issues with the telehealth video coaching (yes/no) and whether the number and duration of the telehealth video coaching sessions was appropriate (just right/preferred more/preferred less).

Appropriateness: participants were asked to rate the relevance of the information presented in the “Body Balance Beyond” website, the coaching sessions and the text messages (e.g., “provided me with useful information about healthy eating”) on a five-point Likert scale (strongly disagree = 1, strongly agree = 5). Participants were also asked to rate the scheduling of the telehealth video coaching calls (“the scheduling of the video coaching sessions was appropriate”).

*Usage:* participants were asked whether they read/saw the AES and Physical Activity reports (3 and 6 months), accessed the “Body Balance Beyond” website (3 and 6 months) or participated in the telehealth video coaching sessions (3 months, high personalisation group only). Attendance rates for the telehealth video coaching sessions were collected objectively (recorded by the dietitian/exercise physiologist) for each participant.

Participants were also asked open-ended questions about what they liked and did not like about each component of the program and were invited to make additional comments about each component and the program overall.

### 2.6. Demographic Characteristics, Pregnancy/GDM History and Health Conditions

Sociodemographic information and data relating to participants’ pregnancy, GDM history and health conditions were collected via questionnaire at baseline. Age, date of birth, country of birth, highest educational qualification completed, Aboriginal/Torres Strait Islander status, marital status, household income, self-reported ability to manage on current household income (scored on a 5-point scale from “impossible” to “easy”) and current smoking status were collected. Participants were asked how many children they had given birth to, the approximate date of their most recent GDM diagnosis, history of GDM in previous pregnancies, GDM management they had received and any other pregnancy-related health conditions.

### 2.7. Statistical Analysis

Questionnaire data were captured using Qualtrics (Provo, UT, USA). Statistical analyses were conducted using SPSS for Windows version 25.0 (SPSS Inc, Chicago, IL, USA). Demographic and baseline characteristics were reported for participants across the three study groups as mean (standard deviation, SD) for continuous variables and as percentages (counts) for categorical variables.

Generalised linear mixed models were used to analyse the preliminary efficacy outcomes for the impact of treatment (high personalisation vs. low personalisation vs. waitlist control), time (baseline, 3 months and 6 months) and the treatment-by-time interaction. Treatment group, time and the treatment × time interaction formed the three terms for the base model, ensuring that outcomes for participants who withdrew prior to 3 months or 6 months were retained in analyses as consistent with an “intention-to-treat” approach. Base models were initially tested using compound symmetry and unstructured variance types, and the appropriate variance type (i.e., yielding the lowest Akaike’s information criterion) was selected for each model. Age and BMI were also assessed to determine any significant interactions in the models for each outcome. Where a covariate was significant, two-way interactions with time and treatment were also examined, and any significant two-way interactions were adjusted for in the model. When a three-way interaction with the covariate by treatment group-by-time was significant, this three-way interaction and all relevant two-way interactions were adjusted for in the model. Coefficients and *p*-values for the treatment-by-time interaction term were examined to determine the efficacy of the intervention using a significance level of *p* = 0.05 for the primary and all secondary outcomes. In addition, generalised linear mixed models were used to analyse the outcome of weight for the impact of treatment of both intervention groups combined vs. waitlist control.

A backward stepwise multiple regression model evaluated variation in the primary outcome of weight change (%) predicted by group, ability to manage on current income, whether the “Body Balance Beyond” website was accessed at 3–6 months (yes/no), change in ARFS (baseline to 6 months), change in moderate to vigorous physical activity (MVPA) (baseline to 6 months) and change in % energy from core foods (baseline to 6 months). Variables not reaching statistical significance (*p* > 0.2) were removed at each step to yield a set of variables that best predicted the outcome.

In addition to absolute weight loss, we investigated the proportion of completers with (i) any weight loss between baseline and 6 months and (ii) clinically significant weight loss at 6 months (defined as ≥ 5% of baseline weight) [46,47]. Group differences were evaluated using the Pearson chi-square test.

For the process evaluation, frequency of each response option was reported at 3 months and 6 months. Responses for both intervention groups were combined for elements of the program that were common to both groups on account of low group numbers. Free text responses to open-ended questions were reported together with frequency data as relevant.

## 3. Results

### 3.1. Recruitment and Retention

Figure 2 summarises the flow of participants through the study. In total, 47 of the 327 individuals who attempted the screening questionnaire were eligible. Main reasons for exclusion were not completing the screening questionnaire (*n* = 184), being <3 months postpartum (*n* = 15), BMI <25 or >50 kg.m^2^ (*n* = 24), not obtaining medical clearance to participate (*n* = 16) or slow internet speed (*n* = 20). Forty-two women consented and were randomised (high personalisation *n* = 15, low personalisation *n* = 13 and waitlist control *n* = 14), of which 30 (71%) completed the study. Retention at 6 months for the three study groups was 80% for high personalisation, 54% for low personalisation and 79% for waitlist control. Reasons for withdrawal included personal/work commitments (*n* = 4), falling pregnant (*n* = 1), being placed on a strict weight-loss diet (*n* = 1), resources not useful (*n* = 1) and uncontactable (*n* = 5).

### 3.2. Participants

Characteristics of women across the three study groups are reported in Table 1. Mean age was 33.5 ± 4.0 years. The majority of women were born in Australia (95%, *n* = 40), one-third found it difficult some of the time to manage on their current household income (33%, *n* = 14) and one-third reported their general health as fair or poor (33%, *n* = 14). Mean time since GDM diagnosis was 12.8 ± 7.3 months. Mean weight was 87.7 ± 13.4 kg, and waist circumference was 100.0 ± 10.5 cm. Mean fasting plasma glucose was 4.8 ± 0.4 mmol/L, and mean HbA1c was 5.1 ± 0.3% (32.1 ± 3.2 mmol/L). Notably, the following biomarkers collected at baseline were within healthy ranges: HbA1c %, fasting glucose, QUICKI and triglycerides.

### 3.3. Preliminary Efficacy

Table 2 summarises the results of the intention-to-treat analyses for group changes from baseline to 3 months and baseline to 6 months of preliminary efficacy outcomes. No significant group-by-time effects were observed for most outcomes (weight, HbA1c, total cholesterol, triglycerides, LDL cholesterol, waist circumference, BMI, body fat mass, skeletal muscle mass, HOMA2-IR, QUICKI, fasting plasma glucose and insulin, blood pressure, ARFS total score and subscales, % energy from core and non-core foods, AQoL, WEL-SF and EX-SE questionnaires, MVPA, frequency of resistance and pelvic floor training) (all *p* > 0.05). A significant group-by-time effect was observed for HDL cholesterol, favouring the low personalisation group relative to the waitlist control group at 6 months (0.19 (95% confidence interval (CI) 0.01, 0.38) mmol/L, *p* = 0.028). There was no significant group-by-time effect observed for the outcome of weight when both treatment groups were compared against the control group (*p* = 0.137).

For the outcome of weight, a trend favouring the intervention groups was observed at 3 months and 6 months, although the differences among all three groups were not significant (*p* = 0.29). The results from multiple regression modelling found that no variables predicted the outcome of % weight change (*p* > 0.05).

Sixteen women (53% of completers: high personalisation *n* = 8, low personalisation *n* = 3 and waitlist control *n* = 5) lost weight at 6 months. Among those who lost weight, mean weight loss was 3.1 ± 1.9 kg (−4.4 ± 3.3 for the high and low personalisation groups combined). Five women (17% of completers: high personalisation *n* = 2, low personalisation *n* = 2 and waitlist control *n* = 1) lost ≥5% of their baseline body weight at 6 months, with one participant in the high personalisation group losing ≥10% (Figure 3). There was no difference in the proportion of participants losing any weight or losing ≥5% body weight across the study groups (*p* > 0.05).

### 3.4. Acceptability

Appendix A presents the results from the process evaluation questions at 3 months and 6 months for the low personalisation and high personalisation groups combined. Responses to process evaluation questions for the intervention elements that both groups received (Australian Eating Survey and physical activity reports, and the “Body Balance Beyond” website) were similar between the two groups although not evaluated statistically due to the small sample sizes.

#### 3.4.1. Australian Eating Survey and Physical Activity reports

The majority of women found the Australian Eating Survey and Physical Activity reports easy to understand (89% and 94% at 3 months and 88% and 94% at 6 months, respectively). The reports helped most women identify areas in their diet and physical activity to improve (95% and 89% at 3 months and 100% and 88% at 6 months, respectively) and set diet and physical activity goals (79% and 78% at 3 months and 82% and 76% at 6 months, respectively). One participant commented that the reports “would have been a good option to complete each month voluntarily to keep [them] top of mind”.

#### 3.4.2. “Body Balance Beyond” Website (High and Low Personalisation Groups)

The majority (91%) of women accessed the “Body Balance Beyond” website at least once in the first 3 months; however, this fell to 57% at 3–6 months. While the website provided useful information about diabetes risk for most women (95% and 92% at 3 and 6 months, respectively) and was easy to navigate (90% and 75% at 3 and 6 months, respectively), only a third of women found it motivating (30% and 25% at 3 and 6 months, respectively). The goal-setting module was used at least once per month by 85% of participants in the first 3 months and 50% of participants in months 3–6. At 3 months, the majority of women agreed that the goal-setting module was easy to use for setting weight, nutrition and exercise goals (82%, 88% and 94%, respectively) (“feeling accountable” and “helped [me to] stay focused”). However, at 6 months, only half (58%) of women were satisfied with how the website supported them in managing their diabetes risk or in developing a plan. (“[the website was] difficult and time consuming to review goals…. I just gave up and haven’t reviewed goals”).

#### 3.4.3. Telehealth Video Coaching Sessions (Months 1–3, High Personalisation Group Only)

All women (100%) agreed that the telehealth video coaching sessions with the dietitian and the exercise physiologist provided useful information in the first 3 months. Ability to engage was high, as most women agreed that the telehealth video coaching increased their confidence with improving diet (85%) and physical activity (92%) behaviours. Half the women would have preferred more contact with the dietitian (54%) or exercise physiologist (46%) than was provided (3 sessions for each) (“more would have helped me stay motivated and accountable”). Most women (85%) were satisfied with the duration of the sessions (20–30 min). Almost half (46%) the women experienced technology issues that delayed or prevented them from completing a session, and analysis of the free text responses indicated that issues arose on account of the Scopia© platform used (“sometimes the Scopia system needed to be updated prior to linking up which was annoying when trying to log on”). Most participants (92%) agreed the telehealth video coaching sessions were easier than attending in person (“not needing to travel or get childcare to attend appointments”), with two-thirds (69%) satisfied with the telehealth video coaching overall. Participants also appreciated the person-to-person contact (“personalised, accountability, friendly and caring” and “the ladies were lovely and did not judge”).

Attendance rates for the dietitian and exercise physiologist coaching sessions were both 100% (*n* = 15) for session 1 and 93% (*n =* 14) for session 2. Attendance for session 3 was lower: 80% (*n =* 12) for the dietitian and 60% (*n* = 9) for the exercise physiologist.

#### 3.4.4. Text Message Support (Months 3–6, High Personalisation Group Only)

Most women (84%) agreed that the text messages provided useful information (“I like that they were gentle reminders to keep me accountable”), although the focus on weight outcomes was a theme identified in the free text responses to aspects of the texts that participants did not like (*“they were very weight related”*). The ability of the text messages to engage was low: agreement rates for “increased my confidence to improve my [diet/activity]” were 22% and 31%, respectively, and “helped me achieve my [weight/nutrition/exercise] goals” were 8%, 15% and 8%, respectively. Less than half (39%) the women were satisfied with the text messages overall (“They just didn’t work for me. I preferred talking to the people in person.”). One participant suggested to “add extra motivational texts with little quotes and facts” to improve the appropriateness of the text messages.

#### 3.4.5. “Body Balance Beyond” Program Overall (High Personalisation Group Only)

Overall, most women in the high personalisation group (82% at 3 months and 87% at 6 months) indicated they would recommend the program to other women with GDM. At 3 months, the program met the expectations of more than half (59%) the participants, although this fell to 46% at 6 months. Two participants (9%) reported being dissatisfied with the program at both time points (Appendix A).

## 4. Discussion

This pilot study identified that the “Body Balance Beyond” healthy lifestyle website combined with telehealth coaching was largely acceptable and useful for women with a previous diagnosis of GDM, although further refinement of the website and text messaging elements could improve outcomes. We found no statistically significant effect on weight change and therefore diabetes risk reduction at the group level. However, sixteen women across the sample (53% of completers) lost weight at 6 months, including five women (17% of completers: high personalisation *n* = 2, low personalisation *n* = 2 and waitlist control *n* = 1) who lost ≥5% of their baseline body weight. Given the positive reception to the program by participants, a larger study with a focus on sustained engagement is warranted to evaluate the efficacy of the “Body Balance Beyond” program in women with a recent history of GDM.

### 4.1. Recruitment and Retention

Recruiting participants for this study was difficult. Of the 327 individuals who attempted the initial screening questionnaire, 184 did not complete the questionnaire and a further 96 were excluded. The most common reasons for exclusion were BMI <25 or >50 kg.m^2^ (*n* = 24), slow internet speed (*n* = 20) or not obtaining medical clearance to participate (*n* = 16). An additional 15 women who were identified as potential participants were <3 months postpartum and, as such, were ineligible for the current study; a possible solution for future studies could be to engage these women to commence the program at 3 months postpartum. The low uptake from the large number initially interested highlights the challenges of recruiting this specific cohort of women.

Although this pilot study was planned as an eHealth trial which was to be delivered remotely, participants were still required to attend the research facility in person at three time points in order to complete objective assessments. A key modification for a future study could be to only include self-reported measures that can be collected via survey. This modification would potentially increase the reach to participants by not restricting recruitment to a particular geographical area and to potentially increase uptake in the study by removing the barriers (e.g., transport and childcare arrangements) associated with attending in-person assessments for this population. Identifying potential participants at the point of diagnosis of GDM and linking the intervention to their antenatal pathway with the support of primary care clinicians is another recruitment strategy worth considering. Wilkinson et al. [48] highlighted the need for improved coordination in the care of women with GDM during their pregnancy and postpartum, noting interactions with the general practitioners as a “missed opportunity”, particularly given that general practitioners feel under-utilised with regard to GDM management [49].

Retention was a further challenge in this pilot study, as also reported by others. The 6-month retention rate of 71% is at the lower end of previous lifestyle intervention studies among women with recent GDM (69–91%) [14,16,22,50]. The high attrition rate in our study could reflect ongoing issues with sustained technology use compounding the known poor retention rates for lifestyle intervention studies. The major barriers to completing the study were reported as personal/work commitments and subsequent pregnancy, similar to earlier studies [14,16,22,50]. Dropout was highest for the low personalisation group, possibly reflecting dissatisfaction with this intervention arm which was self-directed only. Across the sample, only one participant withdrew due to not finding the resources useful (low personalisation group), although a further five women were not contactable so their reasons remain unknown. Factors that could have increased retention in the low personalisation group include clear communication at the start of the study about what the different groups would receive (both in the recruitment materials and the screening questionnaire) and the acceptability of the website element of the program. These form key considerations for the engagement of participants for a future study.

### 4.2. Preliminary Efficacy

From the intention-to-treat analysis, there was no group-by-time effect on any of the preliminary efficacy outcomes except for HDL cholesterol, where a difference favoured the low personalisation group relative to control. This was not unexpected given that this pilot study was not powered for statistical significance. Furthermore, as many of the biomarkers collected at baseline were within healthy ranges (HbA1c %, fasting glucose, QUICKI and triglycerides), there was limited capacity for change in these measures. Other modifiable factors that confer risk for developing type 2 diabetes mellitus, such as smoking, sedentary behaviour and stress/depression, were not evaluated in the current pilot study and remain an area for future investigation.

For the outcome of weight, a trend favouring the intervention groups was observed at 3 months and 6 months. Across the sample, five women, including four women from the intervention groups, achieved clinically significant weight loss (5% or more of their baseline body weight) at 6 months [46,49]. A systematic review of dietary/lifestyle interventions for weight loss in adults found that mean weight loss at 2–3 years was 3.5 ± 2.4 kg, compared to the combined intervention group mean weight loss of 1.5 ± 4.5 kg in the current study. Additionally, 17% of completers in our study achieved ≥5% weight loss, although none of the included studies in the prior systematic reported on the proportion of participants achieving this level of clinically significant weight loss [51]. It is possible that providing more support to women, such as through greater frequency of telehealth video coaching and more personalisation of text messages (as per the feedback from participants) could have improved preliminary efficacy outcomes for women enrolled in the “Body Balance Beyond” program in the current study. Of interest, seven women in the intervention arms (four in the high personalisation (HP) group and three in the low personalisation (LP) group) recorded a gain in weight at the end of the 6 months, compared to four women in the waitlist control group. It is unclear, why these women gained weight as this was not an aspect evaluated in the study; however, such a finding does indicate that exploration of negative outcomes should be considered in future studies. Further, another factor possibly affecting study outcomes was the time since GDM diagnosis. Women were considered eligible if they were between 3–24 months postpartum, and we recognise that the variation in the time since GDM diagnosis observed across the sample could have affected the outcome of weight loss. This remains an area for consideration in future interventions.

### 4.3. Acceptability

The website elements, including the goal-Setting module and the AES and physical activity reports, of the “Body Balance Beyond” program were rated favourably by participants in both the high and low personalisation groups for appropriateness of content and usability. However, satisfaction with and usage of the website at 6 months were lower than anticipated. This could reflect the fact that the website was static, as all content was made available at the start of participants’ enrolment in the study. Modifications that could be introduced to sustain engagement with the website element could include regular text messages and emails to prompt participants to complete certain tasks associated with the website.

Feedback relating to the telehealth video coaching element of the intervention was generally positive in terms of satisfaction and appropriateness. Most women felt that the telehealth video coaching increased their confidence and helped them achieve goals. Furthermore, half the women would have preferred more contact with the dietitian or exercise physiologist (54% and 46%, respectively), indicating the value that participants placed on these sessions. Although only participants who had sufficient internet access were included as per the selection criteria, technology difficulties were still experienced by almost half the participants, which affected usability of this element. The technology issues were identified as relating primarily to the Scopia© platform used to deliver the coaching sessions, highlighting a key area of consideration for future studies.

Satisfaction with the text message support provided from 3–6 months was not rated highly by participants. Ability to engage was low; less than half the women felt that the text messages helped them to eat more healthily, to be more active or to achieve their goals. Although the intention was to individually tailor the text messages for each participant, less than half the women felt that the text messages were personalised, highlighting an area for improvement. The focus on weight in the text messages was not received favourably, suggesting that, while weight loss may be a key goal of the program, focusing on other outcomes, such as reinforcing the importance of positive lifestyle changes, could be more appropriate. Another study used text messages to provide feedback about participants’ progress towards their goals in a 13-week pedometer program for women with recent GDM, although the frequency and level of personalisation of these text messages was not clearly described [22]. Text messages have been used to support self-management among individuals with type 1 and type 2 diabetes mellitus, yielding improvements in weekly physical activity and consumption of fruit and vegetables [52,53]. These prior studies provided a range of information via text messages, such as educational messages, reminders, encouragement, self-assessments and feedback [52,53], providing further potential areas for improvement on the text messaging element of the “Body Balance Beyond” program. In future studies, the texting element of the program could be modified to increase the frequency of text messages and to diversify the content to include information and links to resources as well as to prompt reflection on goals and self-monitoring.

Women with a history of GDM experience substantial barriers to adopting healthy lifestyle behaviours, including carer responsibilities, poor timing of education and fragmented care as well as additional cultural barriers for women born overseas [54,55]. While intensive lifestyle interventions are effective at delaying or preventing progression to diabetes [12], such interventions require extensive resources and support. Lifestyle programs aimed at reducing future diabetes risk in women with previous GDM delivered using eHealth technologies are considered highly acceptable by the women themselves, along with health professionals [19]. Although some modifications to elements of the “Body Balance Beyond” website are likely needed to improve sustained engagement (for example, providing more interactive, personalized content), the current study provides further evidence on the acceptability of eHealth programs among women with a history of GDM.

## 5. Conclusions

Recruiting and retaining women with a recent diagnosis of GDM is challenging. The “Body Balance Beyond” website combined with telehealth video coaching appears to be acceptable for women with recent GDM, although further refinement of the website and text messaging elements is suggested. Further analysis of the program’s efficacy on diabetes risk reduction in a larger study is warranted.

## Figures and Tables

**Figure 1 ijerph-17-07115-f001:**
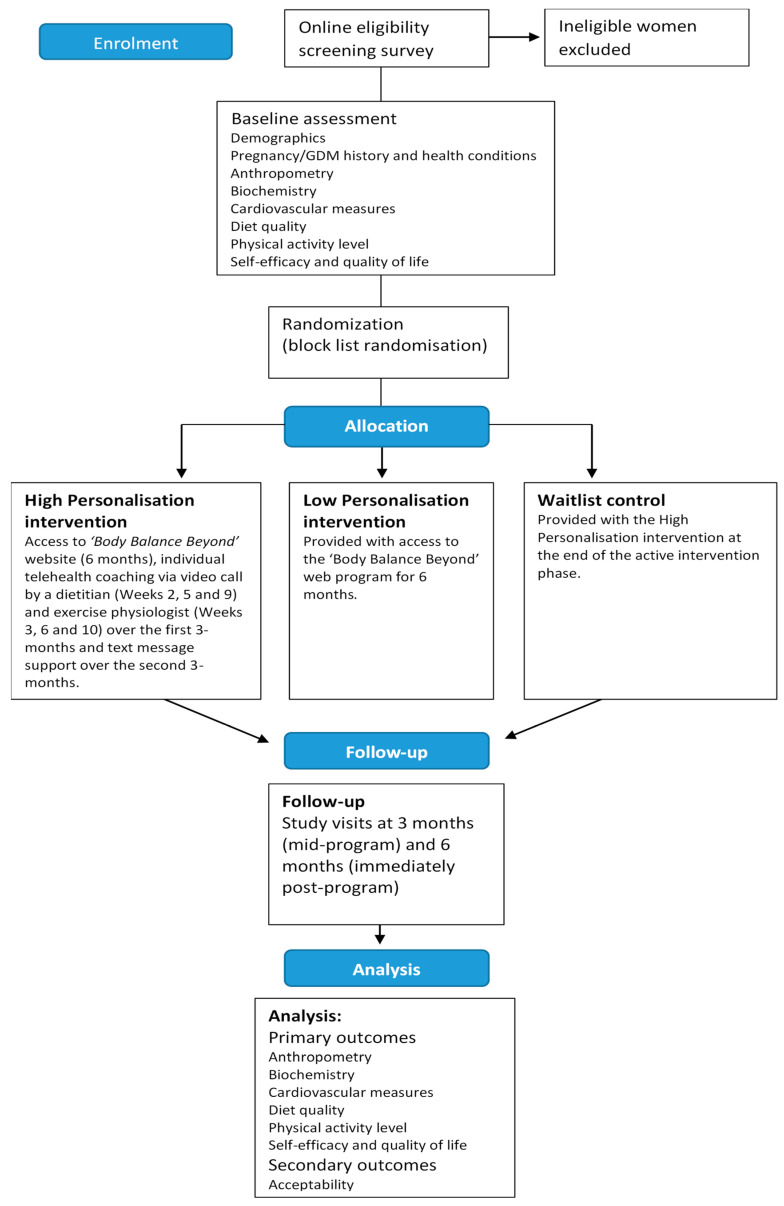
Study protocol for a 6-month pilot randomised controlled trial for women with recent gestational diabetes mellitus (GDM) who were at risk of developing Type 2 diabetes mellitus (T2DM).

**Figure 2 ijerph-17-07115-f002:**
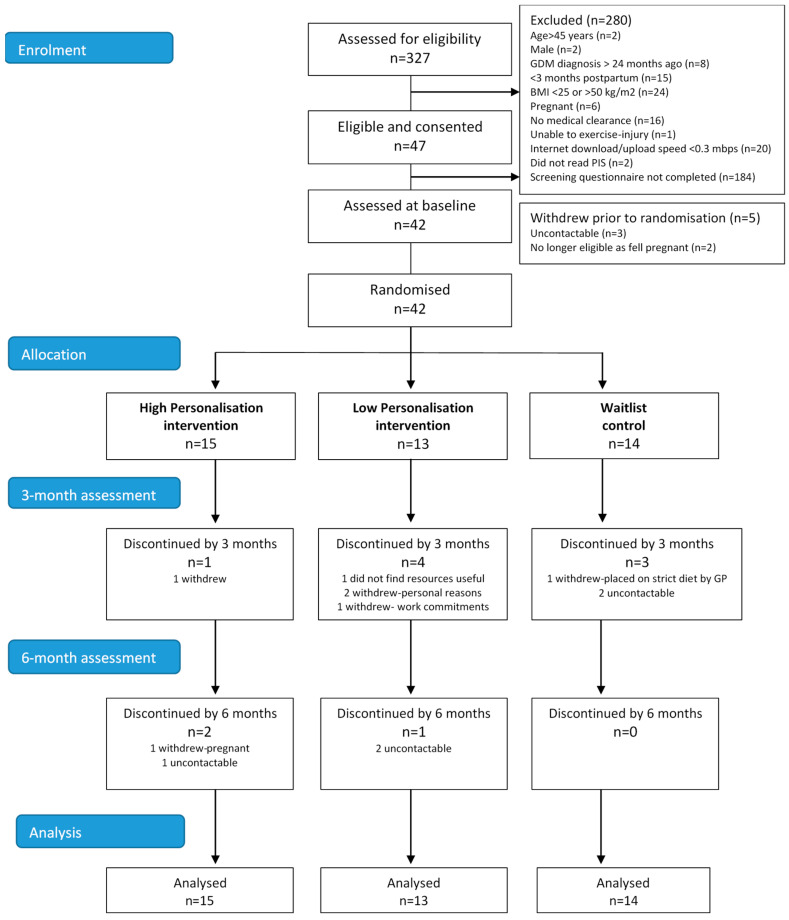
Consolidated Standards of Reporting Trials (CONSORT) diagram showing the flow of participants in a 6-month pilot randomised controlled trial for women with recent GDM who were at risk of developing T2DM.

**Figure 3 ijerph-17-07115-f003:**
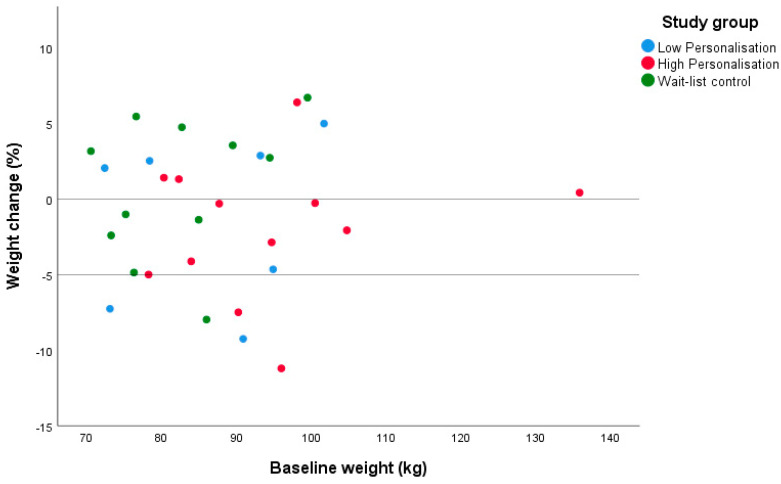
Individual weight change (% of baseline weight) at 6 months by group in a pilot randomised controlled trial for women with recent gestational diabetes who were at risk of developing type 2 diabetes mellitus.

**Table 1 ijerph-17-07115-t001:** Baseline characteristics of women randomised to the high personalisation, low personalisation and waitlist control groups in a 6-month pilot randomised controlled trial for women with recent GDM who were at risk of developing T2DM: data are presented as % (n) or mean ± SD.

	High Personalisation	Low Personalisation	Waitlist Control
	(*n* = 15)	(*n* = 13)	(*n* = 14)
**Sociodemographic Characteristics**			
Age	34.0 ± 4.5	32.8 ± 3.6	33.6 ± 3.8
Country of birth			
Australia	93.3 (14)	100 (13)	92.9 (13)
Highest qualification completed			
School certificate (year 10 or equivalent)	6.7 (1)	0	14.3 (2)
Certificate/diploma/trade	26.7 (4)	30.8 (4)	35.7 (5)
University degree	60.0 (9)	69.2 (9)	50 (7)
Marital status			
Married/de facto	93.3 (14)	100 (13)	100 (14)
Never married	6.7 (1)	0	0
Household income			
<$1000 weekly	6.7 (1)	0	7.1 (1)
$1000–$1999 weekly	46.7 (7)	38.5 (5)	35.7 (5)
>$2000 weekly	33.3 (5)	53.8 (7)	57.1 (8)
Do not know/decline to answer	13.3 (2)	7.7 (1)	0
Ability to manage on current income			
Difficult some of the time	46.7 (7)	15.4 (2)	35.7 (5)
Not too bad/easy	53.4 (8)	84.6 (11)	64.3 (9)
Current smoker (less than once per week)	0	7.7 (1)	0
**Pregnancy and GDM History**			
Parity	2.1 ± 1.2	1.6 ± 1.0	1.7 ± 0.6
Months since first GDM diagnosis	14.4 ± 10.1	11.6 ± 5.6	12.2 ± 4.9
**GDM Management**			
Diet	93.3 (14)	92.3 (12)	71.4 (10)
Exercise	40.0 (6)	46.2 (6)	28.6 (4)
Tablets	0	7.7 (1)	0
Insulin	66.7 (10)	46.2 (6)	64.3 (9)
**Other Health Conditions**			
Pre-eclampsia	20.0 (3)	23.1 (3)	7.1 (1)
PCOS	20.0 (3)	23.1 (3)	7.1 (1)
Low thyroid hormone levels	6.7 (1)	0	7.1 (1)
Other condition: cholestasis of pregnancy	0	7.7 (1)	0
**Anthropometry**			
Height (m)	1.66 ± 0.10	1.62 ± 0.10	1.65 ± 5.60
Weight (kg)	91.1 ± 15.9	89.3 ± 12.5	83.8 ± 10.7
BMI (kg/m^2^)	32.8 ± 4.1	33.9 ± 3.6	31.1 ± 4.8
BMI category, % (n)			
Overweight	26.7 (4)	23.1 (3)	28.6 (4)
Obese	73.3 (11)	76.9 (10)	57.1 (8)
Waist circumference (cm)	101.5 ± 12.2	103.0 ± 10.6	96.3 ± 8.1
Body fat mass (kg)	35.7 ± 9.6	39.1 ± 10.2	40.4 ± 11.2
Skeletal muscle mass (kg)	26.7 ± 3.2	27.3 ± 2.9	28.1 ± 3.8
**Biochemistry**			
HbA1c %	5.1 ± 0.3	5.1 ± 0.4	5.1 ± 0.3
Fasting blood glucose (mmol/L)	4.8 ± 0.3	4.7 ± 0.5	4.9 ± 0.4
Fasting insulin (mU/L)	7.8 ± 3.0	10.3 ± 5.0	6.1 ± 2.6
HOMA2-IR	1.0 ± 0.4	1.3 ± 0.6	0.8 ± 0.3
QUICKI	0.36 ± 0.02	0.35 ± 0.03	0.37 ± 0.03
LDL cholesterol (mmol/L)	3.4 ± 1.2	3.2 ± 0.7	3.7 ± 1.0
HDL cholesterol (mmol/L)	1.4 ± 0.3	1.3 ± 0.3	1.5 ± 0.3
Total cholesterol/HDL ratio	3.8 ± 0.8	4.2 ± 1.1	4.1 ± 1.2
Triglycerides (mmol/L)	1.1 ± 0.4	1.2 ± 0.7	1.2 ± 0.5
**Cardiovascular Measures**			
Systolic blood pressure (mmHg)	104.4 ± 8.2	109.4 ± 9.4	106.4 ± 10.2
Diastolic blood pressure (mmHg)	67.5 ± 6.1	69.3 ± 5.6	68.6 ± 7.3
**Dietary Intake**			
ARFS total score (maximum 73)	34.7 ± 6.4	39.0 ± 10.3	34.2 ± 7.0
% energy: core foods	57.4 ± 11.1	62.9 ± 7.0	60.9 ± 12.6
% energy: non-core foods	42.6 ± 11.1	37.1 ± 7.0	39.1 ± 12.6
% energy: protein	18.7 ± 2.8	21.3 ± 3.0	18.6 ± 3.2
% energy: carbohydrate	45.7 ± 4.7	39.7 ± 4.2	43.9 ± 6.6
% energy: fats	35.1 ± 3.6	38.8 ± 3.7	33.6 ± 14.4
% energy: saturated fats	15.5 ± 2.4	17.4 ± 2.6	14.1 ± 1.8
% energy: alcohol	1.3 ± 2.2	1.0 ± 1.6	4.1 ± 5.0
**Physical Activity**			
MVPA (minutes/week)	74.3 ± 87.1	144.2 ± 114.4	130.0 ± 116.7
Resistance training frequency			
None	86.7 (13)	69.2 (9)	57.1 (8)
1–2 times per week	6.7 (1)	23.1 (3)	7.1 (1)
3 or more times per week	6.7 (1)	7.7 (1)	35.7 (5)
Pelvic floor exercise frequency			
None	53.3 (8)	69.2 (0)	42.9 (6)
1–2 times per week	13.3 (2)	15.4 (2)	35.7 (5)
3 or more times per week	20.0 (5)	15.4 (2)	21.4 (3)
**Self-Efficacy and Quality of Life**			
WEL-SF (maximum 90)	43.9 ± 19.7	51.5 ± 16.5	46.6 ± 14.4
Self-Efficacy for Exercise score (maximum 11)	5.9 ± 1.9	5.8 ± 2.2	6.0 ± 2.5
AQoL-6D utility score (maximum 1.0)	0.76 ± 0.11	0.78 ± 0.11	0.80 ± 0.20

Abbreviations: AQoL, Assessment of Quality of Life 6-dimension; ARFS, Australian Recommended Food Score; BMI, body mass index; GDM, gestational diabetes mellitus; HbA1c, glycated haemoglobin; HDL, high-density lipoprotein; HOMA-IR2, homeostatic model assessment of insulin resistance; LDL low-density lipoprotein, MVPA, moderate to vigorous physical activity; PCOS, polycystic ovarian syndrome; QUICKI, quantitative insulin sensitivity check index; SD, standard deviation; and WEL-SF, Weight Efficacy Lifestyle Questionnaire-Short Form.

**Table 2 ijerph-17-07115-t002:** Mean (95% CI) change in preliminary efficacy outcomes within groups and between groups (intention-to-treat population) over time in a 6-month pilot randomised controlled trial for women with recent GDM who were at risk of developing T2DM.

		Change from Baseline, Mean (95% CI) ^a^			Difference between Groups, Mean (95% CI) ^b^			*p*-Value
Outcome	Month	High Personalisation	Low Personalisation	Waitlist Control	High Personalisation vs. Waitlist Control	Low Personalisation vs. Waitlist Control	High vs. Low Personalisation	
		*n* = 15	*n* = 13	*n* = 14				
Weight (kg) ^c^	3	−1.30 (−0.50, 3.10)	−0.91 (−3.15, 1.32)	1.11 (−1.08, 3.29)	−2.41 (−5.24, 0.42)	−2.02 (−5.15, 1.11)	−0.39 (−3.26, 2.48)	0.391
	6	−1.60 (−3.50, 0.31)	−0.90 (−3.36, 1.57)	0.75 (−1.27, 2.78)	−2.35 (−5.13, 0.43)	−1.65 (−4.84, 1.54)	−0.70 (−3.81, 0.43)	
HbA1c (%)	3	0.02 (−0.07, 0.11)	0.04 (−0.08, 0.15)	0.09 (−0.02, 0.20)	−0.07 (−0.21, 0.07)	−0.05 (−0.21, 0.10)	−0.01 (−0.15, 0.13)	0.673
	6	0.06 (−0.03, 0.16)	0.02 (−0.10, 0.14)	0.11 (0.01, 0.21)	−0.05 (−0.19, 0.09)	−0.09 (−0.25, 0.07)	0.04 (−0.11, 0.20)	
Total cholesterol (mmol/L)	3	−0.34 (−0.74, 0.06)	−0.37 (−0.86, 0.12)	−0.02 (−0.50, 0.47)	−0.32 (−0.95, 0.31)	−0.35 (−1.04, 0.34)	0.03 (−0.61, 0.67)	0.769
	6	−0.30 (−0.72, 0.13)	−0.38 (−0.92, 0.16)	−0.34 (−0.79, 0.11)	0.04 (−0.58, 0.66)	−0.04 (−0.74, 0.67)	0.08 (−0.61, 0.77)	
HDL cholesterol (mmol/L)	3	−0.09 (−0.19, 0.02)	0.02 (−0.11, 0.15)	0.03 (−0.10, 0.16)	−0.12 (−0.28, 0.05)	−0.01 (−0.19, 0.17)	−0.10 (−0.27, 0.06)	**0.028**
	6	−0.11 (−0.22, 0.01)	−0.06 (−0.20, 0.08)	−0.25 (−0.37, −0.13)	0.14 (−0.02, 0.31)	**0.19 (0.01, 0.38)**	−0.05 (−0.23, 0.13)	
Triglycerides (mmol/L) ^d^	3	−0.02 (−0.20, 0.15)	0.01 (−0.21, 0.22)	−0.07 (−0.28, 0.14)	0.04 (−0.23, 0.32)	0.08 (−0.23, 0.38)	−0.03 (−0.31, 0.25)	0.091
	6	−0.14 (−0.33, 0.04)	−0.01 (−0.25, 0.23)	0.18 (−0.02, 0.37)	−0.06 (−0.23, 0.11)	−0.14 (−0.32, 0.06)	0.07 (−0.11, 0.26)	
ARFS (maximum 73)	3	0.87 (−2.24, 3.98)	−1.08 (−4.91, 2.74)	−1.90 (−5.50, 1.71)	2.77 (−2.03, 7.57)	0.81 (−4.48, 6.11)	1.95 (−3.01, 6.92)	0.274
	6	1.79 (−1.41, 4.99)	2.89 (−1.10, 6.89)	−2.00 (−5.48, 1.47)	3.79 (−0.97, 8.55)	4.89 (−0.44, 10.23)	−1.11 (−6.26, 4.05)	
% energy: core ^d^	3	10.48 (5.00, 15.96)	6.11 (−0.45, 12.68)	0.72 (−5.76, 7.19)	9.73 (1.26, 18.20)	4.87 (−4.36, 14.10)	4.86 (−3.68, 13.40)	0.143
	6	8.51 (1.31, 15.71)	8.51 (1.31, 15.71)	3.12 (−2.91, 9.14)	9.29 (0.96, 17.62)	4.99 (−4.41, 14.39)	4.30 (−4.92, 13.53)	
% energy: non-core ^d^	3	−10.48 (−5.00, −15.96)	−6.11 (−12.68, 0.45)	−0.72 (−7.19, 5.76)	−9.73 (−18.20, −1.26)	−4.87 (−14.10, 4.36)	−4.86 (−13.40, 3.68)	0.143
	6	−8.51 (−1.31, −15.71)	−8.51 (−15.71, 1.31)	−3.12 (−9.14, 2.91)	−9.29 (−17.62, −0.96)	−4.99 (−14.39, 4.41)	−4.30 (−13.53, 4.92)	
MVPA (min/week) ^d^	3	60.17 (−10.67, 131.01)	−21.68 (−108.05, 64.69)	−21.30 (−107.25, 64.65)	82.06 (−173.44, 337.56)	11.32 (−264.43, 287.08)	70.74 (−186.78, 328.26)	0.158
	6	182 (−35, 400)	−78 (−360, 230)	−42 (−269, 232)	244 (−8, 496)	52 (−228, 333)	191 (−84, 467)	

Abbreviations: AQoL = Assessment of Quality of Life 6-dimension; ARFS = Australian Recommended Food Score; BMI = body mass index; CI = confidence interval; GDM = gestational diabetes mellitus; HbA1c = glycated haemoglobin; HDL = high-density lipoprotein; HOMA-IR2 = homeostatic model assessment of insulin resistance; LDL = low-density lipoprotein; MVPA = moderate to vigorous physical activity; QUICKI = quantitative insulin sensitivity check index; SEE = Self-Efficacy for Exercise; T2DM = type 2 diabetes mellitus; and WEL-SF = Weight Efficacy Lifestyle Questionnaire-short form. ^a^ Time differences were calculated as 3 months minus baseline and 6 months minus baseline. ^b^ Between-group differences in changes from baseline to 6 months. ^c^ Adjusted for age. ^d^ Adjusted for BMI. Significant p-values (*p* < 0.05) are indicated in bold.

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
