# Peer review of "The Feasibility and Preliminary Efficacy of an eHealth Lifestyle Program in Women with Recent Gestational Diabetes Mellitus: A Pilot Study"

_ijerph, 2020, doi:10.3390/ijerph17197115_

Round 1
Reviewer 1 Report
Many thanks for the opportunity to review this feasibility and efficacy study investigating an eHealth program for women who experienced GDM in their pregnancy.
This was a very well designed program that used valid measures and was thoughtfully and rigorously delivered.
It is important to publish, despite the limited clinical changes observed due to the contribution it makes to the literature regarding program content (and the process outcomes presented) as well as (continuing) to highlight the difficulties recruiting women of this population group so we can all learn from what's been tried.
I suggest minor changes and questions that will only serve to further strengthen the paper or future research design and delivery.
INTRODUCTION:
line 64 - are there other examples to justify this statement (also can be used in Discussion).
E.g. O'Reilly et al (2016) the MAGDA study publication and Peacock et al (2014) the "WENDY" / walking for exercise and nutrition to prevent diabetes in you study.
Was telephone counselling or the website the main focus of the promotion of the program? The website is presented first but I wonder if this takes second fiddle to the counselling aspect (with the personalisation and tailoring aspect) that is then backed up through targeted promo and links to parts of the website?
METHODS
Even though you state this wasn't powered to detect the changes in the primary outcome later in the results (4.2 Prelim efficacy) you state this wasn't sufficiently powered. It would be interesting to present the numbers you WOULD have needed (esp with the trend in wt loss).
Line 124; line 163 you note the texts were 'tailored'; it'd be great to have a couple of examples of these
Lines 339-340 - ? do you think the HP and LP groups had different expectations of the website due to what else they did/didn't receive
RESULTS
?were any data collected on the waitlist group after they did the HP
p9 ?had any women developed T2DM who were recruited
3.4.3 - interesting that women requested more visits/contacts; this is supported by the NHMRC owt/ob guidelines. Perhaps contacts similar to this frequency would assist bedding down momentum for behaviour change?
DISCUSSION
line 490 - I may be misreading this but I'm curious as to why this should start at 3 months? or recruit from 3 months? I was going to note why not build on previously established relationships WITHIN GDM services and recruit then, to deliver postnatally. I also see you mention this later with bringing in GPs which is great (see Wilkinson et al MJA 2014 Who is responsible...?)
It is extremely difficult to demonstrate the efficacy of a low intensity intervention when requiring hospital visits for pathology and anthropometric measures. Is there a partnership with local/community pathology units that can be forged to assist in some data collection? Or at least the GPs?
As noted above, might be good to bring in O'Reilly and Peacock to support the statements you've made regarding recruitment, retention, outcome data and intervention's design.
Additionally, Willcox et al's txt4two (2017) study is a good example of building tailored, responsive text messages to great effect.
Other suggestions drawn from these studies and extending your ideas could be 'group telehealth'?
Good luck with future program delivery.
Author Response
Response to reviewers
Reviewer 1
Comments and Suggestions for Authors
Many thanks for the opportunity to review this feasibility and efficacy study investigating an eHealth program for women who experienced GDM in their pregnancy.
This was a very well designed program that used valid measures and was thoughtfully and rigorously delivered.
It is important to publish, despite the limited clinical changes observed due to the contribution it makes to the literature regarding program content (and the process outcomes presented) as well as (continuing) to highlight the difficulties recruiting women of this population group so we can all learn from what's been tried.
I suggest minor changes and questions that will only serve to further strengthen the paper or future research design and delivery.
INTRODUCTION:
1.) line 64 - are there other examples to justify this statement (also can be used in Discussion).
E.g. O'Reilly et al (2016) the MAGDA study publication and Peacock et al (2014) the "WENDY" / walking for exercise and nutrition to prevent diabetes in you study.
Author response:
We have added two additional references to support the statement at Line 66 as follows
- O’Reilly et al. Mothers after Gestational Diabetes in Australia (MAGDA): A Randomised Controlled Trial of a Postnatal Diabetes Prevention Program. Plos Medicne. 2016 Jul 26;13(7):e1002092.
- Peacock et al. A Randomised Controlled Trial to Delay or Prevent Type 2 Diabetes after Gestational Diabetes: Walking for Exercise and Nutrition to Prevent Diabetes for You. Int J Endocrinol. 2015;2015:423717.
2.) Was telephone counselling or the website the main focus of the promotion of the program? The website is presented first but I wonder if this takes second fiddle to the counselling aspect (with the personalisation and tailoring aspect) that is then backed up through targeted promo and links to parts of the?
Author response:
The website component of the intervention was the focus of the recruitment materials. In these materials, it was also stated that participants may also have access to individualised coaching sessions with a dietitian and exercise physiologist delivered via video call. The participant information statement specified that participants could be in one of the three treatment arms.
METHODS
3.) Even though you state this wasn't powered to detect the changes in the primary outcome later in the results (4.2 Prelim efficacy) you state this wasn't sufficiently powered. It would be interesting to present the numbers you WOULD have needed (esp with the trend in wt loss).
Author response:
Preliminary calculation indicates that the sample size per group is likely to vary between 200 up to 500 per group, depending on the comparison made. Given there would potentially be changes in a future study design we feel it is better to not include a sample size calculation. Hence, we have not altered the text.
4.) Line 124; line 163 you note the texts were 'tailored'; it'd be great to have a couple of examples of these
Author response:
Example text messages have been added to the manuscript at lines 171-179, as follows:
‘When completing the second personalized nutrition and exercise questionnaires, participants prioritized their top 3 nutrition and top 3 exercise factors and four messages for each factor sent over 12 weeks. An example text message for the nutrition capability factor of ‘having the ability to resist food cravings’ was “[First name], when cravings hit, have a glass of water and wait a few mins. You may be confusing thirst for hunger. The craving may fade once you have re-hydrated.” An example text message for the exercise opportunity factor of “Having better access to exercise, physical activity facilities or equipment” was “[First name], Exercise equipment is all around you. Here is a park bench circuit [LINK] for you to try, with step ups, tricep dips and many more!”.’
5.) Lines 339-340 - ? do you think the HP and LP groups had different expectations of the website due to what else they did/didn't receive
Author response:
This is quite possible, however this was not assessed. The use of self-directed materials, as a standalone intervention, may not be appropriate for all individuals. It is likely that the one-on-one support provided by the dietitian and exercise physiologist may have made use of the website more personalised for those in the HP group.
RESULTS
6.) ?were any data collected on the waitlist group after they did the HP
Author response:
No data was collected on this group following their completion of the HP intervention.
7.) p9 ?had any women developed T2DM who were recruited
Author response:
A diagnosis of T2DM was an exclusion criteria. This has been clarified in the manuscript at lines 108-111 as follows:- .
‘Exclusion criteria were: currently pregnant or trying to fall pregnant (includes women who became pregnant during the course of the study); no medical clearance to exercise; medical condition or injury that could be exacerbated by exercise; diagnosed with Type 1 or Type 2 diabetes mellitus…’
8.) 3.4.3 - interesting that women requested more visits/contacts; this is supported by the NHMRC owt/ob guidelines. Perhaps contacts similar to this frequency would assist bedding down momentum for behaviour change?
Author response:
We agree and the current study supports flexibility of choice in regard to tailoring the amount of support an individual may or may not need and benefit from.
DISCUSSION
9.) line 490 - I may be misreading this but I'm curious as to why this should start at 3 months? or recruit from 3 months? I was going to note why not build on previously established relationships WITHIN GDM services and recruit then, to deliver postnatally. I also see you mention this later with bringing in GPs which is great (see Wilkinson et al MJA 2014 Who is responsible...?)
Author response:
The decision to recruit women from 3 months post-partum was based on another study we conducted in overweight/obese women (Vincze et al. [1]). Similarly to this earlier study, we wanted to allow sufficient time for recovery post-birth for participating women, however we acknowledge that earlier engagement with the target sample may have improved enrolment.
We have expanded this point in the discussion at lines 526-528:
‘Wilkinson et al.,[2] highlight the need for improved coordination in the care of women with GDM during their pregnancy and post-partum, noting interactions with the general practitioners as a ‘missed opportunity.’
10.) It is extremely difficult to demonstrate the efficacy of a low intensity intervention when requiring hospital visits for pathology and anthropometric measures. Is there a partnership with local/community pathology units that can be forged to assist in some data collection? Or at least the GPs?
Author response:
We agree and this is a challenge of demonstrating impact on clinical outcomes in an initial pilot study. We believe this data can inform interpretation of self-report only in future trials.
11.) As noted above, might be good to bring in O'Reilly and Peacock to support the statements you've made regarding recruitment, retention, outcome data and intervention's design.
Author response:
Reference to these studies in relation to recruitment and retention has been noted in the discussion at lines 530-532 and lines 534-536.
12.) Other suggestions drawn from these studies and extending your ideas could be 'group telehealth'?
Author response:
Thank you for the suggestion. We will consider this option if opportunity for future intervention delivery arises.
Reviewer 2 Report
This is a well-conducted and thoroughly-described study addressing an important area: reducing weight and type 2 diabetes risk in women with a recent history of gestational diabetes. Efficacy results are largely null, but this is not surprising in a pilot/feasibility study wherein mean biomarkers were in the normal range at baseline. The main message of the work is that eHealth interventions are feasible in this group, yet reach is low, recruitment challenged, and engagement over time suboptimal. I offer a few points for the authors to consider.
How was the program developed? Were potential participants or other stakeholders consulted? Why did the authors think these specific intervention components would be acceptable and effective?
Rationale is not provided for the high vs low personalization arms. I presume the goal was to compare a higher cost versus lower cost option, which has implications for sustainability. Could the authors give an indication of staff burden and total costs between the two interventions? They note the scalability of the intervention is a strength, but the personalization adds implementation burden that works against sustainability.
A figure depicting the protocol (timing of intervention components and data collection) would be helpful.
Table 3 is overwhelming. It would be easier to understand as a bar chart (one bar per question per timepoint) with different shades indicating the proportion agreeing, neutral, or disagreeing.
A table summarizing the key barriers and potential approaches to address them in future studies would enhance readability and accessibility.
Author Response
Reviewer 2
Comments and Suggestions for Authors
This is a well-conducted and thoroughly-described study addressing an important area: reducing weight and type 2 diabetes risk in women with a recent history of gestational diabetes. Efficacy results are largely null, but this is not surprising in a pilot/feasibility study wherein mean biomarkers were in the normal range at baseline. The main message of the work is that eHealth interventions are feasible in this group, yet reach is low, recruitment challenged, and engagement over time suboptimal. I offer a few points for the authors to consider.
1.) How was the program developed? Were potential participants or other stakeholders consulted? Why did the authors think these specific intervention components would be acceptable and effective?
Author response:
Learnings from our previous research [1, 3, 4] was used to guide the selection of online delivery including the addition of video calls with a dietitian and exercise physiologist. In these studies, the delivery of an eHealth intervention and use of video calls were aspects that postpartum women indicated they would want from a weight management program. Other aspects were developed specifically for this study and/or adapted for post-partum women (e.g. personalised nutrition and exercise questionnaires).
2.) Rationale is not provided for the high vs low personalization arms. I presume the goal was to compare a higher cost versus lower cost option, which has implications for sustainability. Could the authors give an indication of staff burden and total costs between the two interventions? They note the scalability of the intervention is a strength, but the personalization adds implementation burden that works against sustainability.
Author response:
The intention of the two personalisation arms was to determine if a lower level of personalisation intervention had a different effect compared to a higher level of personalisation through the addition of one-on-one health professional and text message support. Obviously, the delivery of the high personalisation required more resources to implement due to the nature of the additional components involved. However, a cost comparison between intervention arms was not an aim of the study and therefore will not be calculated.
We agree that stating the program is scalable is an overstatement. This sentence has now been removed.
3.) A figure depicting the protocol (timing of intervention components and data collection) would be helpful.
Author response:
Thank you for this suggestion. We have now provided a figure depicting the protocol.
Please refer to Figure 1.
4.) Table 3 is overwhelming. It would be easier to understand as a bar chart (one bar per question per timepoint) with different shades indicating the proportion agreeing, neutral, or disagreeing.
Author response:
In line with the recommendation by reviewer 3, Table 3 has now been moved to supplementary material.
5.) A table summarizing the key barriers and potential approaches to address them in future studies would enhance readability and accessibility.
Author response:
We thank the reviewer for this suggestion, however we feel the use of sub-headings in the discussion summarise the key themes and the subsequent text concisely summarises the approaches that could be used in future studies.
Reviewer 3 Report
Overall a good pilot study. Some minor comments below:
There are several primary outcomes. I think section 2.4 should read primary outcomes.
Section 2.4 mentions that participants were blinded. The blinding seems impossible to me. Please explain how the blinding was ensured.
There is recall bias throughout the study. For example, in line 240 asked about physical activity over the past month. It will be good to mention the “recall bias” as a limitation.
Line 244 mentions that vigorous physical activity was calculated for minutes per week. How was this calculated?
Line 302: approximate date of the most recent GDM diagnosis – recall bias again.
Line 309: should report median and IQR instead of mean and SD whenever the data is skewed. Does the BMI is symmetric in all groups?
Line 331: in the backward elimination p > 0.05 is very strong. It can be used for interaction terms but for eliminating variables AICC or BIC or a p > 0.2 might be a reasonable choice.
Page 9: wait-list or waitlist – be consistent
Table 1: Only reports current smoker. All others are past smokers or non-smokers or current smoker with more than once per week?
Do your analysis taken into account for GDM management? In Table 1 insulin was used by 66.7% of the high personalisation group. This is a confounding variable. Did your models adjusted for this or taken it into account?
Figure 2: should use the word weight change instead of weight loss. If it is weight change then negative means loss and positive means gain. There are three (red dots) high personalisation gained weight. One of them was more than 5%. This needs to be discussed and explained the underlying reasons.
The article is very long. Please move Table 3 to an appendix.
Line 505: Is this overall retention rate? The retention rate was different at the beginning of the article.
Author Response
Reviewer 3
Comments and Suggestions for Authors
Overall a good pilot study. Some minor comments below:
1.) There are several primary outcomes. I think section 2.4 should read primary outcomes.
Authors response:
Despite being a pilot study, weight change was the primary outcome. This has now been clarified in the manuscript at line 198.
2.) Section 2.4 mentions that participants were blinded. The blinding seems impossible to me. Please explain how the blinding was ensured.
Author response:
The manuscript states that: “Participants were blinded to group allocation until after their baseline assessment”. We appreciate this may be confusing for some readers so we have removed this sentence.
3.) There is recall bias throughout the study. For example, in line 240 asked about physical activity over the past month. It will be good to mention the “recall bias” as a limitation.
Author response:
We acknowledge that when asked to report behaviours retrospectively, there is potential for recall bias to impact on accuracy of reporting. However, validated questionnaires were used for the key behaviours which is important to highlight.
4.) Line 244 mentions that vigorous physical activity was calculated for minutes per week. How was this calculated?
Authors response:
Participants reported the duration (in minutes) and frequency (per week) of light, moderate and vigorous intensity physical activity they engaged in the past month using a questionnaire. We summed total times for moderate to vigorous physical activity which is indicated in lines 260-261, as follows:
‘The total times for moderate and vigorous activities were summed to provide a measure of moderate to vigorous physical activity (minutes/week).’
5.) Line 302: approximate date of the most recent GDM diagnosis – recall bias again.
Author response:
See earlier comment. We mention that this data is self-reported. The inclusion of this data is to describe to the reader the characteristics of the sample.
6.) Line 309: should report median and IQR instead of mean and SD whenever the data is skewed. Does the BMI is symmetric in all groups?
Author response:
Data was not significantly skewed and therefore mean and SD was chosen for reporting.
7.) Line 331: in the backward elimination p > 0.05 is very strong. It can be used for interaction terms but for eliminating variables AICC or BIC or a p > 0.2 might be a reasonable choice.
Author response:
Sorry the p value was incorrectly reported. We have now corrected this error at line 349:
‘Variables not reaching statistical significance (p > 0.2)….’
8.) Page 9: wait-list or waitlist – be consistent
Authors response:
Thank you for identifying this. We have now removed the hyphen from waitlist in figure 2.
9.) Table 1: Only reports current smoker. All others are past smokers or non-smokers or current smoker with more than once per week?
Author response:
This information is provided to give an overall summary of characteristics of participating women, therefore current smoking status was reported.
10.) Do your analysis taken into account for GDM management? In Table 1 insulin was used by 66.7% of the high personalisation group. This is a confounding variable. Did your models adjusted for this or taken it into account?
Author response:
Women in the study are not currently pregnant and therefore were no longer being managed for GDM, and nor do they have type 2 diabetes and hence we feel inclusion of this variable as a cofounder is not needed. As per earlier comment, this information is provided to give an overall summary of characteristics of women choosing to participate.
11.) Figure 2: should use the word weight change instead of weight loss. If it is weight change then negative means loss and positive means gain. There are three (red dots) high personalisation gained weight. One of them was more than 5%. This needs to be discussed and explained the underlying reasons.
Author response:
The figure axis label has been changed.
In addition, the following text has been added to the discussion lines 564-568:
“Of interest, seven women in the intervention arms (four in HP group and three in LP group) recorded a gain in weight at the end of the 6 months, compared to four women in the waitlist control group. It is unclear, why these women gained weight as this was not as aspect evaluated in the study, however such a finding does indicate that exploration of negative outcomes should be considered in future studies
12.) The article is very long. Please move Table 3 to an appendix.
Authors response:
Thank you for this suggestion. We have now moved Table 3 to the Supplementary material.
13.) Line 505: Is this overall retention rate? The retention rate was different at the beginning of the article.
Author response:
Yes, 71% was the overall retention rate. Both the abstract and line in the discussion are the same. The abstract also includes the retention rates for each intervention arm.